# Multi-Omics Analysis Reveals Age-Related Microbial and Metabolite Alterations in Non-Human Primates

**DOI:** 10.3390/microorganisms11102406

**Published:** 2023-09-26

**Authors:** Xiang Chen, Yiyun Liu, Juncai Pu, Siwen Gui, Dongfang Wang, Xiaogang Zhong, Wei Tao, Xiaopeng Chen, Weiyi Chen, Yue Chen, Renjie Qiao, Peng Xie

**Affiliations:** 1Department of Neurology, The First Affiliated Hospital of Chongqing Medical University, Chongqing 400016, China; chenxiang202303@163.com (X.C.);; 2NHC Key Laboratory of Diagnosis and Treatment on Brain Functional Diseases, The First Affiliated Hospital of Chongqing Medical University, Chongqing 400016, China

**Keywords:** aging, non-human primates, systems biology, gut microbiome, serum and fecal metabolome

## Abstract

Aging is a systemic physiological degenerative process, with alterations in gut microbiota and host metabolism. However, due to the interference of multiple confounding factors, aging-associated molecular characteristics have not been elucidated completely. Therefore, based on 16S ribosomal RNA (rRNA) gene sequencing and non-targeted metabolomic detection, our study systematically analyzed the composition and function of the gut microbiome, serum, and fecal metabolome of 36 male rhesus monkeys spanning from 3 to 26 years old, which completely covers juvenile, adult, and old stages. We observed significant correlations between 41 gut genera and age. Moreover, 86 fecal and 49 serum metabolites exhibited significant age-related correlations, primarily categorized into lipids and lipid-like molecules, organic oxygen compounds, organic acids and derivatives, and organoheterocyclic compounds. Further results suggested that aging is associated with significant downregulation of various amino acids constituting proteins, elevation of lipids, particularly saturated fatty acids, and steroids. Additionally, age-dependent changes were observed in multiple immune-regulatory molecules, antioxidant stress metabolites, and neurotransmitters. Notably, multiple age-dependent genera showed strong correlations in these changes. Together, our results provided new evidence for changing characteristics of gut microbes and host metabolism during aging. However, more research is needed in the future to verify our findings.

## 1. Introduction

Aging is a multifactorial process of inevitable decline in physiological functions of humans and other organisms, involving various systems of the body, with the nervous system being the primary focus [1]. During the process, individual interacting and interdependent organs throughout the body exhibit distinct biological characteristics and are associated with increased risk of disease [2]. Previous studies have indicated that neurodegenerative diseases such as Alzheimer’s disease, Parkinson’s disease, and amyotrophic lateral sclerosis are related to aging [3,4,5]. The systems biology research on microbiome and metabolome changes during aging has become a focal point. However, a comprehensive understanding of molecular alterations is still needed [6,7].

16S rRNA gene sequencing, which amplifies and sequences specific regions of bacterial 16S rRNA genes, rapidly and efficiently identifies microbial species in the gut microbiome. The technique provides a crucial tool for investigating the intimate relationship between the gut microbiome and host health, as well as the mechanisms behind various diseases [8]. However, there is still a need to elucidate the dynamic changes in gut microbiota during the aging process using 16S rRNA gene sequencing. The gut microbiome plays a crucial role in various physiological processes, such as host energy metabolism, immune regulation, and signal transduction [9]. Previous research indicated that the human gut microbiome undergoes dynamic changes with age. For example, *Bifidobacterium* significantly increases during the neonatal period [10], while Proteobacteria and Actinobacteria abundance rises between 3 to 6 months after birth [11]. In adulthood, the gut microbiome is primarily composed of Bacteroidetes and Firmicutes [12], but in old individuals, there is a higher proportion of Bacteroidetes and reduced levels of beneficial genera such as *Bifidobacterium* and *Lactobacillus* [13,14].

However, the microbiota patterns of human individuals are highly variable, influenced by factors such as environment, diet, and medication usage [15,16]. Rodents are the most widely used laboratory animals in existing gut microbiome studies and have important scientific value in the field [17]. However, non-human primates (NHPs) share a high degree of similarity with humans in terms of genetics, anatomy, reproduction, development, cognition, and social structure [18]. Studies of NHPs can be designed to minimize the effects of confounding factors seen in human-based research, making them ideal models to characterize gut microbiome changes during aging [19]. However, research using NHPs as aging models remains limited and varies in age grouping and analytical methods. Therefore, further exploration of age-dependent composition and function altered of gut microbiome based on NHPs is important to the field.

The aging process involves extensive changes in metabolism, but systematic studies based on serum and fecal metabolome are still scarce. Evidence in recent years suggested that aging was associated with progressive metabolic changes. For instance, early studies identified significant correlations between multiple metabolites and age in the cortical tissues spanning from infants to the elderly [20]. Rodent research also revealed age-dependent metabolites in peripheral blood [21]. Notably, recent evidence indicated that gut microbiota may mediate changes in the phenotype of aging mice through the regulation of host physiological metabolism [22]. However, holistic metabolic changes and molecular interrelationships during aging remain to be elucidated. Therefore, integrating the serum and fecal metabolic profile of NHPs holds significant importance for exploring the patterns of metabolic changes during aging.

To date, few studies have systematically integrated the gut microbiome, fecal, and serum metabolic profile of NHPs at different ages, leading to a need for further clarification of the interrelationship among them. In this study, the microbiome and metabolome of male rhesus macaques at three stages, namely juvenile, adult, and oldness, were analyzed by 16S rRNA gene sequencing and non-targeted metabolomic technologies. This study will contribute to elucidating the characteristics and interrelationships of the gut microbiota and host metabolism, and provides new evidence for a series of molecular changes accompanied by aging.

## 2. Materials and Methods

### 2.1. Inclusion of Rhesus Macaques and Ethics Statement

Rhesus macaques were housed in a standard monkey house of Zhongke Experimental Animal Co., Ltd. (Suzhou, Jiangsu, China), with free access to drinking water, and regular feeding of compound high-nutrition food twice daily along with fresh fruits or vegetables once daily. Based on prior research [23,24,25], the rhesus macaques were categorized into three groups including juvenile (1–4 years), adult (7–15 years), and oldness (≥16 years). Additionally, stringent health criteria were applied, including veterinary diagnosis and quarantine records to ensure the recruited rhesus macaques were free from specific pathogens. The included rhesus macaques had no record of drug use for six months.

This study was approved by Chongqing Medical University and followed the “Guide for the Care and Use of Laboratory Animals” of the Institute of Neuroscience, Chongqing Medical University. The work involving non-human primates adhered to both the NIH guide for the care and use of laboratory animals and the recommendations of the Weatherall report. We also followed nc3r recommendations (https://www.nc3rs.org.uk/) and accessed on 1 March 2021.

### 2.2. Collection of Serum and Fecal Samples

Recruited rhesus macaques were anesthetized by intramuscular injection of pentobarbital (30 mg/kg, concentration 4%), and 3 mL of peripheral blood was collected through the brachial vein using procoagulant tubes. The blood samples were allowed to stand at room temperature for 30 min, and then centrifuged at 3000 rpm for 15 min at room temperature. The supernatants were transferred to sterile EP tubes and stored at −80 °C in a refrigerator [26]. Then, the rhesus macaques were placed in metabolic cages, and feces samples were collected for 2 days using a sterile feces sampler. In order to reduce the degradation of microbial DNA, the feces excreted by each rhesus macaque were collected within 1 h. The middle part of the sample was intercepted by the core cutting method, packed in a 1.5 mL sterile EP tube, placed in an anaerobic bag, and immediately stored at −80 °C [24].

### 2.3. DNA Extraction, PCR Amplification, and Illumina Sequencing

Following manufacturer instructions of the PF Mag-Bind Stool DNA Kit (Omega Bio-tek, Norcross, GA, USA), we extracted microbial genomic DNA. PCR targeted the 16S rRNA gene V3–V4 region using 338F (5’-ACTCCTACGGGAGGCAGCAG-3’) and 806R (5’-GGACTACHVGGGTWTCTAAT-3’) primers under the following conditions: initial denaturation at 95 °C for 3 min, followed by 27 cycles of denaturing at 95 °C for 30 s, annealing at 55 °C for 30 s, and extension at 72 °C for 45 s, and single extension at 72 °C for 10 min, and end at 4 °C [27]. Each sample had 3 replicates, pooled and purified from 2% agarose gel, and size-verified. Quantification was carried out using Promega’s Quantus™ Fluorometer (Promega, Madison, WI, USA). Purified PCR products underwent Bioo Scientific’s NEXTFLEX Rapid DNA-Seq Kit library prep (Austin, TX, USA). Illumina’s PE300/PE250 platforms (Illumina, San Diego, CA, USA) were sequenced by Shanghai Meiji Biomedical Technology Co., Ltd. (Shanghai, China) Raw data were submitted to the NCBI SRA database.

### 2.4. 16S rRNA Gene Sequence Analysis

The paired-end raw sequencing reads were quality-controlled using the fastp software (version 0.19.6) and subsequently merged using the FLASH software (version 1.2.11) [28]. With default parameters, the quality-controlled merged sequences were denoised using the DADA2 plugin from the Qiime2 pipeline (version 2020.2). Sequences post DADA2 denoising are referred to as ASVs (Amplicon Sequence Variants) [29]. Chloroplast and mitochondrial sequences were removed from all samples. To mitigate the impact of sequencing depth on subsequent alpha and beta diversity analyses, all sample sequences were rarefied to 20,000. Even after rarefaction, the average sequence coverage per sample remained at 99.09%. ASVs were taxonomically classified using the Qiime2 Naive Bayes classifier based on the Sliva 16S rRNA gene database (version 138). Alpha diversity indices such as Chao and Shannon were calculated using mothur software (version 1.30.2) [30], a partial least-squares discriminant analysis (PLS-DA) was used to explore the differences and similarities of microbial compositions among the three groups [31]. Linear discriminant analysis effect size analysis (LEfSe) was conducted to identify differentially abundant bacterial taxa among different age groups (LDA > 2, *p* < 0.05) [32].

### 2.5. Serum and Fecal Metabolomics Analysis

This protocol was consistent with our previous research, with minor modifications [33,34]. We collected 50 μL serum samples mixed with 200 μL ice-cold water–methanol–chloroform (2:5:2, *v*/*v*) and 20 μL internal standard. After 5 s vortexing, samples underwent 30 min ultrasonication. Drying occurred at 37 °C with nitrogen. MeOX pyridine solution (50 μL) was incubated at 37 °C for 90 min, followed by the addition of 60 μL BSTFA and 2 h incubation at 37 °C. The supernatant was used for gas chromatography-mass spectrometry analysis (GC-MS; 8890GC-5977B MSD, DB-5). The methodology for GC-MS detection of fecal samples is detailed in our previous study [34]. PLS-DA was performed on normalized data. The variable importance in projection (VIP) was computed via the PLS-DA model [35]. Based on ANOVA analysis, the overall *p*-values among the metabolites of three groups were calculated, followed by post-hoc tests between pairs of groups using the least significant difference (LSD) method. Only metabolites meeting the criteria of *p* < 0.05 and VIP > 1 were considered to exhibit statistically significant differences [36].

### 2.6. Statistical and Bioinformatics Analysis

Statistical analyses were carried out using SPSS version 21.0 (SPSS, Chicago, IL, USA). For metabolomic data, logarithmic transformations were applied for normalization. Subsequently, ANOVA was performed followed by LSD’s multiple comparison analysis to assess differences among groups, only metabolites meeting the criteria of *p* < 0.05 and VIP > 1 were considered to exhibit statistically significant differences [36]. Metabolites or gut genera showing significant differences between any two groups were subjected to Spearman correlation analysis to determine their association with age. A correlation was deemed significant when *p* < 0.05 [37].

Possible pathways affected by the altered gut microbiota were predicted through the application of Phylogenetic Investigation of Communities by Reconstruction of Unobserved States (PICRUSt) analysis, based on the Kyoto Encyclopedia of Genes and Genomes (KEGG) database. Subsequently, a LEfSe analysis was employed to identify significantly different biological pathways (LDA > 2 and *p* < 0.05) [38,39]. The MetaboAnalyst was used to conduct pathway analysis for metabolites with significant differences between two groups and a significant correlation with age. The statistical method was the hypergeometric test, and the background library was based on the KEGG database. A change in the pathway was considered significant when *p* < 0.05 [40].

## 3. Results

### 3.1. Overall Characteristic of the Enrolled Rhesus Monkeys

A total of 36 male rhesus macaques, age from 3 to 26 years were recruited. Following previous research, they were categorized into three age groups: juvenile (9 individuals), adult (12 individuals), and oldness (15 individuals). All of them were subjected to the same dietary and housing conditions. The detailed characteristics of the enrolled rhesus macaques are presented in Appendix A.

### 3.2. Age-Related Alterations in Gut Microbiome

To resolve the composition and functional characteristics of the gut microbiome at different ages, we used 16S rRNA gene sequencing to compare the relative abundance of gut microbiota in juvenile, adult, and old rhesus macaques. At the phylum level, the gut microbiota of three groups were dominated by Firmicutes and Bacteroidetes (Appendix A). At the family level, *Lachnospiraceae*, *Lactobacillaceae*, *Streptococcaceae*, and *Spirillaceae* accounted for a relatively high proportion (Figure 1A). In the α-diversity analysis, Kruskal–Wallis tests revealed significant overall differences in Chao (*p* = 0.007), and Shannon (*p* = 0.041) indices among the three groups and significant differences were observed the juvenile and old groups (Figure 1B).

To investigate the compositional differences of the gut microbiome, β-diversity analysis was conducted among juvenile, adult, and old macaques. PLS-DA score plots at the ASV level showed distinct clustering of gut microbiome among the three groups (R2X = 0.199, R2Y = 0.585, Q2 = 0.180; Figure 1C). Furthermore, LEfSe analysis was utilized to identify significant differences in gut microbial taxa among different age groups. The results indicated that 74 taxa at the genus level exhibited significant differences between at least two age groups (LDA > 2, *p* < 0.05; Appendix A). To explore the correlation between differential genera and age, we screened 32 genera with positive correlation with age based on the Spearman’s correlation analysis with the thresholds of *p* < 0.05 and R > |0.3|, such as *Mogibacterium*, *Eubacterium_oxidoreducens_group*, *Oribacterium*, *Terrisporobacter*, *Ruminococcus*, etc. Nine differential genera showed negative correlation with age, such as *Lactobacillus*, *Haldemannia*, *and Coprococcus* (Figure 1D and Appendix A).

To predict the age-dependent functional characterization of gut microbes, 16S rRNA microbiome data were mapped to pathways using PiCrust2 (version 2.2.0). The results indicated that compared with the juvenile group, the four biological processes of neurodegeneration, aging, cofactor and vitamin metabolism, and cell activity were significantly enriched in the adult and old, while amino acid metabolism, environmental adaptation mechanism, secondary metabolite biosynthesis, and prokaryotic cell community biosynthesis were only significantly enriched in the old. Furthermore, glucose metabolism, lipid metabolism, biodegradation and metabolism of exogenous substances, nucleotide metabolism, and body repair mechanism were significantly enriched in the juvenile compared with the old (Figure 1E). Together, our results suggested that there may be significant differences in the composition and functions of gut microbiome across age groups.

### 3.3. Age-Related Changes in Fecal Metabolic Profile

The fecal metabolome can be used to characterize gut microbiota functions and their interactions with the host [41]. Therefore, an analysis of the fecal metabolome was conducted. PLS-DA revealed distinct separation among the fecal metabolite profiles across ages (Figure 2A). We further identified 151 differential metabolites between at least two age groups (*p* < 0.05, VIP > 1; Appendix A), with 37 showing a negative correlation and 49 exhibiting a positive correlation with age (*p* < 0.05; Appendix A). These age-related differential metabolites are mainly distributed in four classes, including lipids and lipid-like molecules, organic oxygen compounds, organic acids and derivatives, and organoheterocyclic compounds (Figure 2B and Appendix A). Furthermore, we identified seven significantly enriched pathways (*p* < 0.05), among which the top-ranked metabolic pathways were aminoacyl-tRNA biosynthesis, pantothenate and CoA biosynthesis, and steroid biosynthesis (Figure 2C). Together, notable changes in various molecules during aging, especially metabolites involving amino acids and lipids molecules, were indicated by our fecal metabolic profile.

### 3.4. Age-Related Changes in Serum Metabolome

To further reveal the overall characteristics of host metabolism during aging, we analyzed differential metabolites in peripheral blood by untargeted metabolomics technology. The PLS-DA analysis revealed significant discrimination of serum metabolite profiles among juvenile, adult, and oldness (Figure 3A). A total of 75 differential metabolites were identified, showing significant differences between at least two age groups (*p* < 0.05, VIP > 1; Appendix A). Correlation analysis revealed 49 metabolites significantly associated with age, of which 20 metabolites increased while 29 metabolites decreased with age (*p* < 0.05; Figure 3C). Consistent with the results of the fecal metabolome, these metabolites mainly belonged to four classes, namely organic acids and derivatives, lipids and lipid-like molecules, organic oxygen compounds, and organoheterocyclic compounds (Figure 3B,E). MetaboAnalyst was used to analyze age-correlated metabolites in the serum and identified 10 significantly enriched pathways. The top-ranked pathways included aminoacyl-tRNA biosynthesis, biosynthesis of unsaturated fatty acids, and D-glutamine and D-glutamate metabolism (Figure 3D). Together, consistent with the fecal metabolome, we found that multiple molecules involved in amino acid, lipid, and lipid-like metabolism were significantly altered in serum across age.

### 3.5. Correlation Analysis of Age-Dependent Factors

To investigate the association among host metabolites during aging, we analyzed correlations between age-dependent serum and fecal metabolites. The results suggest that the correlations between serum and fecal metabolites of the same category were mainly positive, while different categories of metabolites showed complex interrelationships with each other (Figure 4B). To further investigate the interplay between gut microbiota and host metabolism, we performed Spearman correlation analyses between age-correlated gut genera and four major classes of age-related metabolites in serum and feces. The results revealed that most of the metabolites in serum organic acids and their derivatives were significantly correlated with four bacterial genera (*Oribacterium, Solobacterium, Eubacterium*, *and Libanicoccus*). Most of the metabolites in serum lipids and lipid-like molecules showed significant correlations with three bacterial genera (*Catenibacterium, Solobacterium*, *and Holdemanella*). Serum organic oxygen compounds primarily correlated with the *Catenibacterium*, while serum organic heterocyclic compounds mainly correlated with four bacterial genera (*Holdemanella, Solobacterium, Phascolarctobacterium, and Oribacterium*). Regarding fecal samples, most metabolites from the four major classes showed significant correlations with five bacterial genera (*Eubacterium_oxidoreducens_group, Desulfovibrio, Lachnospiraceae_NK3A20_group, Phascolarctobacterium, and Terrisporobacter*) (Figure 4A). These findings suggest that a complex correlation network between gut microbiota and host metabolism may be involved in the aging process.

### 3.6. Co-Occurrence Analysis of Age-Dependent Gut Microbiota with Host Amino Acids and Lipids

From the above results, it can be observed that multiple age-dependent amino acids and lipids in serum and feces are significantly correlated with gut microbiota. To further explore the interaction between the two, strong correlations network diagrams were constructed for amino acids and lipids with bacterial genera using a threshold of *p* < 0.05 and R > |0.5|. The findings reveal a decline in various age-associated protein synthesis-related amino acids with increasing age (Figure 5). Notably, L-tryptophan and L-tyrosine in serum exhibit a robust positive correlation with *Libanicoccus*, while L-methionine and L-serine show a substantial negative correlation with *Escherichia-Shigella*. Similarly, in fecal samples, several protein synthesis-related amino acids display a marked negative correlation with bacterial genera including *Lachnospiraceae_NK3A20_group*, *Eubacterium_oxidoreducens_group*, *Desulfovibrio*, and *Mogibacterium*. Furthermore, *Terrisporobacter* shows a strong positive correlation with diverse fecal lipids such as saturated fatty acids, steroid lipids, and fatty alcohols, while *Coprococcus* and *Holdemanella* exhibit strong positive correlations with multiple fatty acids in serum. Together, our correlation network suggests that various amino acids and lipids in serum and feces may closely interact with gut microbiota during aging.

## 4. Discussion

By integrating 16S rRNA gene sequencing and untargeted metabolomics, we identified 41 microbial taxa at the genus level, along with 86 fecal metabolites and 49 serum metabolites significantly associated with age in non-human primates. Functional analyses revealed alterations in multiple biological processes dominated by amino acid and lipid metabolites, and correlation analysis indicated a complex network between microbiota and host metabolism during aging. These findings provided new evidence for age-dependent changes in gut microbiota and metabolism, offering important insights into the molecular characteristics of aging.

Our multi-omics results indicated age-related differences in amino acid metabolism, with a notable decline in several amino acids crucial for protein synthesis and a decrease in the relative abundance of nutrient-absorbing genera during aging. Above all, levels of L-leucine, L-methionine, L-serine, L-tyrosine, and 4-hydroxyproline in serum showed significant negative correlation with age, in line with previous studies [42]. A key characteristic of aging is the decline in overall protein synthesis [43]. The downregulation of these amino acids may be related to the decline of protein synthesis capacity [44]. Moreover, in feces, we found that protein-synthesizing amino acids such as L-alanine, L-tryptophan, L-alpha-aminobutyric acid, and beta-alanine were negatively correlated with age. This further confirms that amino acid uptake as well as protein synthesis may significantly diminish during the aging process [45]. Our results highlighted a strong positive correlation between *Libanicoccus* and various amino acids composition of proteins in serum. Earlier research documented that *Libanicoccus* has the activities of multiple amino acid metabolic enzymes such as valine arylamidase and leucine arylamidase, and participates in the digestion and absorption of a variety of biological macromolecules [46,47]. This suggests that diminished abundance of *Libanicoccus* may be one of the influencing factors of the age-related decline in protein synthesis capacity. However, the existing studies on *Libanicoccus* are limited and further research is needed to confirm it. Furthermore, our results revealed a robust negative correlation between the *Eubacterium_oxidoreducens_group* and amino acids constituents of proteins in feces. Prior studies noted a significant association between increased *Eubacterium_oxidoreducens_group* and decreased gastric acid secretion [48]. Considering that gastric acid secretion directly influences protein breakdown and amino acid absorption [49], we speculated that with advancing age, the increased abundance of *Eubacterium_oxidoreducens_group* may potentially be in antagonistic relation to protein catabolism and amino acid absorption. However, further research is warranted to confirm this hypothesis. Together, our results imply that the declining levels of amino acids, coupled with the alteration in the abundance of nutrient-absorbing bacterial genera, could be a contributory factor to the diminished protein synthesis capacity during the aging process.

In contrast to amino acid metabolism, our study revealed a positive correlation between most lipids and lipid-like molecules with age, specifically age-related increases in fecal saturated fatty acids, including tridecanoic acid, suberic acid, pentadecanoic acid, and serum stearic acid. Multiple studies have indicated that elevated saturated fatty acids are associated with risks of cardiovascular issues, gallstones, obesity, and metabolic syndrome [50,51,52,53], and heightened saturated fatty acids during aging may indirectly amplify the vulnerability to age-related disorders [54,55]. Notably, our results indicated that *Lachnospiraceae_NK3A20_group* and *Terriporobacter*, which significantly increased during aging, had a strong positive correlation with the aforementioned saturated fatty acids. Early studies reported that the *Lachnospiraceae_NK3A20_group* promotes lipid breakdown and enhances glycerophospholipid digestion and absorption [53,54]. Additionally, *Terriporobacter* has also been implicated in the regulation of bile acid metabolism enzymes and lipid biosynthesis, potentially mediating elevated host lipid levels and disrupted lipid metabolism [56]. However, to date, studies on the regulatory roles of *Lachnospiraceae_NK3A20_group* and *Terrisporobacter* on host lipid metabolism are still limited due to insufficient research. Therefore, studies are needed in the future to elucidate the potential mechanisms and causal effects of these recognized associations. Together, our results indicate significant alterations in lipid metabolism during aging, with the notable features being the increase in abundance of saturated fatty acids and lipid metabolism-related microbial communities.

Notably, our results also suggest that organisms may face more oxidative stress and cognitive decline across age, characterized by a significant decrease in some genera and metabolites with antioxidant and neuroprotective effects. At the metabolic level, as crucial antioxidant and neuroprotective molecules, γ-tocopherol, oleic acid, and linoleic acid in serum were negatively correlated with age. As active forms of vitamin E, multiple studies have shown γ-tocopherol’s ability to neutralize free radicals and reduce oxidative stress, thus providing essential cellular protection [57,58]. A study focusing on the elderly population demonstrated a significant positive correlation between cerebral cortical gamma-tocopherol and synaptic proteins that play a critical role in cognitive function [59]. Multiple studies indicated that the downregulation of γ-tocopherol during aging may be one of the potential factors for cognitive decline [60,61]. Similarly, studies have shown that oleic acid and linoleic acid play an important role as antioxidants and in the improvement of neurodegenerative diseases [62,63]; increased oleic acid intake improves cognitive function in the elderly [64], while higher linoleic acid intake provided neuroprotection and antioxidant stress in a Parkinson’s disease model [65]. Consistently, in the gut microbial dimension, our results suggest that *Coprococcus* and *Lactobacillus* were inversely associated with age. As core genera in the gut, *Coprococcus* and *Lactobacillus* play a crucial role in host resistance to oxidative stress [66,67]. Studies reported that the reduction of *Coprococcus* was associated with the aggravation of gut oxidative stress [68]. Some strains demonstrate potent antioxidant activity through strong scavenging of anions [69]. In conclusion, our results suggest that organisms may be exposed to more oxidative stress and reduced neuroprotective molecules during aging.

In addition, our study further revealed that immune-related factors were significantly altered during aging. Above all, we observed that *Desulfovibrio* and *Escherichia-Shigella* were positively correlated with age. *Desulfovibrio*’s ability to utilize certain fatty acids as a carbon source leads to the production of toxic compounds such as hydrogen sulfide, potentially contributing to the development of inflammatory bowel diseases [70,71]. Previous research in old mice showed an upregulation of *Desulfovibrio*, which subsequently decreased upon treatment with anti-inflammatory drugs, accompanied by improvements in gut inflammatory factors [72]. In addition, *Escherichia*-*Shigella* is another group of bacteria with inflammatory activity [73], which was positively correlated with the levels of IL-1β, NLRP3, and CXCL2 [74]. Regarding the metabolite level, our results indicated positive correlation of serum arachidonic acid with age [75], which served as a precursor for pro-inflammatory mediators such as prostaglandins and leukotrienes [76], and its elevated levels were associated with inflammatory diseases [77]. Taken together, these findings suggest that organisms may be more susceptible to inflammation during aging.

Furthermore, we observed that multiple neurotransmitters and their precursors were significantly altered. Some metabolites that require gut microbiota for biochemical transformation [78], such as L-tryptophan, indole-3-propionic acid, 5-hydroxyindoleacetic acid, and L-dopa, were negatively correlated with age in feces. Similarly, L-tryptophan and L-tyrosine, which is the precursor of L-dopa, were also negatively correlated with age in serum. These metabolites are involved in the synthesis and catabolism of multiple neurotransmitters [79,80], and have potential links to age-related neuropsychiatric disorders [81,82]. Our findings provided further support for earlier studies that microbiota–metabolite pathways in the metabolism of multiple neurotransmitters are crucial in regulating aging phenotypes [83,84].

The study had several limitations: (1) Our sample size was relatively small, pointing to the need for further research with larger cohorts. (2) Due to the limitations of experimental conditions, we were unable to conduct in-depth verification of age-related microbiota and metabolites, urging the focus of future studies to address this aspect. (3) This was a cross-sectional study, while the optimal approach to investigate the influence of age on gut microbiota and host metabolites should be to collect samples longitudinally. However, the dynamic collection of feces and serum over a decade time-span is challenging, and the gut microbiota and host metabolites may be significantly changed under long-term storage conditions. (4) The stratification of the old stage in rhesus macaques has not yet reached consensus, which may be related to complex aging phenotypes observed in non-human primates [85]. Further studies are needed to unify the stratification in the future.

## 5. Conclusions

In the current study, we investigated age-related changes in the gut microbiome, serum, and fecal metabolome in male rhesus macaques. We observed significant correlations between 41 gut genera, 86 fecal, and 49 serum metabolites with age. Our results suggest that aging may be associated with significant downregulation of multiple amino acids constituting proteins, notable elevation of lipids, particularly saturated fatty acids, and steroids. Additionally, age-dependent changes were observed in multiple immune-regulatory molecules, antioxidant stress metabolites, and neurotransmitters. The age-dependent gut microbiota showed a strong correlation in these changes, which indicated that microbe–metabolite interaction networks are involved in the regulation of aging phenotype and homeostasis. Nevertheless, our results warrant further in-depth research for validation.

## Figures and Tables

**Figure 1 microorganisms-11-02406-f001:**
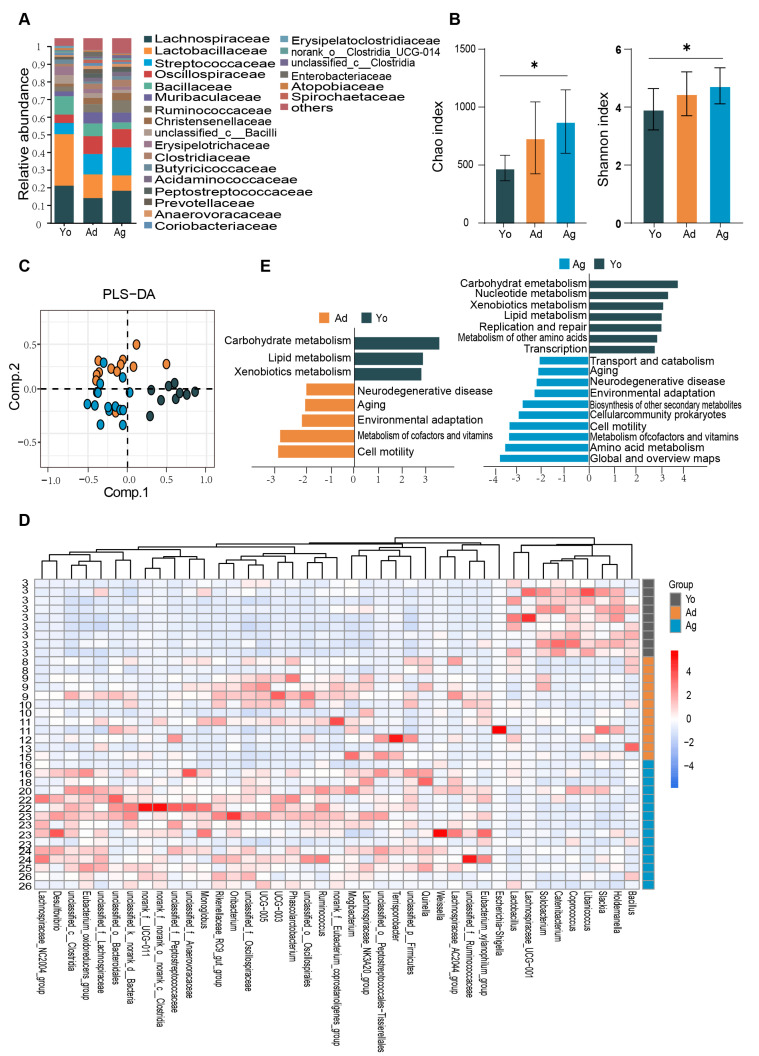
Age−dependent changes in gut microbiome. (**A**) Bar plots showing the relative abundance of microbiota of three groups at family level. (**B**) Alpha−diversity estimates of three groups according to Chao and Shannon indices. Dark gray represents the Juvenile group (Yo), dark yellow represents the Adult group (Ad), and dark blue represents the Old group (Ag). The statistical significance is denoted (* *p* < 0.05). (**C**) PLS−DA score plots of the microbiota among three groups. The color represents different age groups, consistent with (**B**). (**D**) Heatmap showing the relative abundance of three groups at genus level. Only genera with significant differences between at least two groups and correlation with age were displayed (see Appendix A). The numbers on the left represent ages. (**E**) Biological functions predicted by PiCrust2 of gut microbiota across three groups. Differentiating pathways identified by LefSe with *p* < 0.05 and LDA > 2. Yo, juvenile, *n* = 9; Ad, adult, *n* = 12; Ag, oldness, *n* = 15.

**Figure 2 microorganisms-11-02406-f002:**
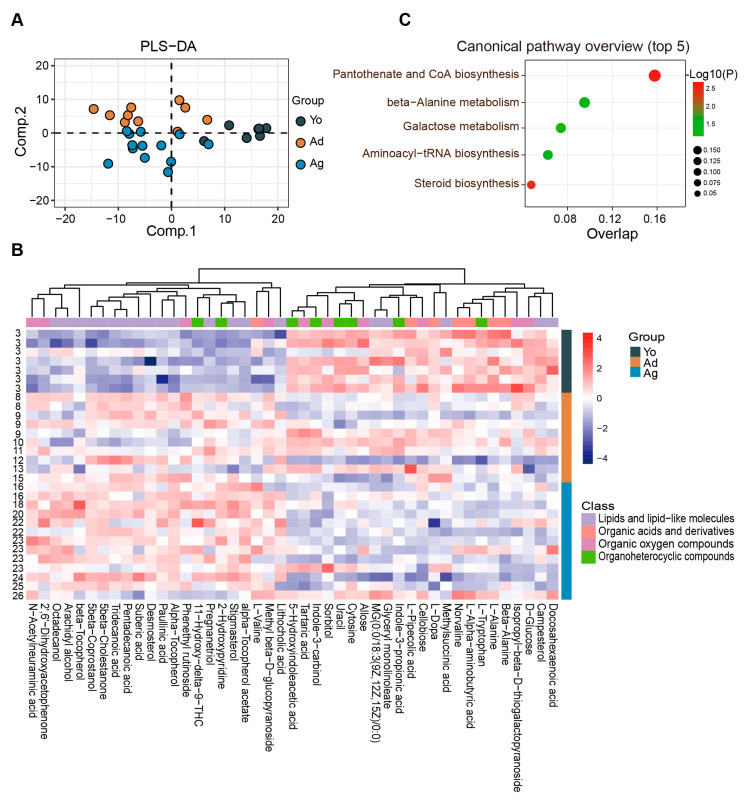
Characteristics of age-related alterations in fecal metabolomic profiles. (**A**) PLS−DA analysis of the metabolome among three groups. (**B**) Heatmap showing the relative abundance of three groups. Only metabolites with significant differences between at least two groups and correlation with age were displayed (see Appendix A). The numbers on the left represent ages. (**C**) The plot for the results of canonical pathway analysis. Nodes represent significantly enriched metabolic pathways. The x axis shows overlap rate of numbers actually matched from the user-uploaded metabolites and the total number of molecules in the pathways. The y axis shows the pathway. Yo, juvenile, *n* = 7; Ad, adult, *n* = 10; Ag, oldness, *n* = 13.

**Figure 3 microorganisms-11-02406-f003:**
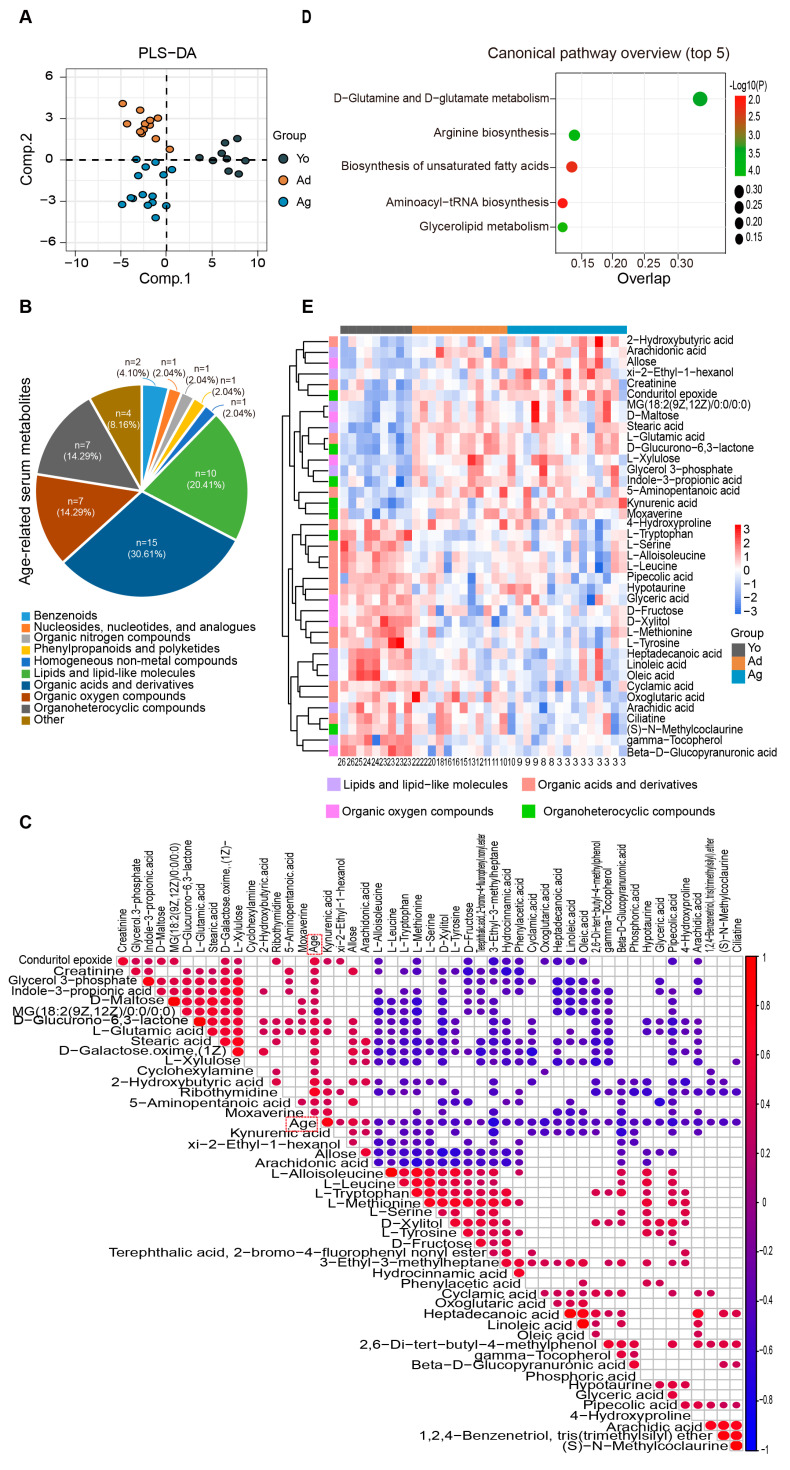
Alterations in serum metabolome associated with age. (**A**) PLS−DA analysis of the metabolome among three age groups. (**B**) Consistent with the fecal metabolome, the pie chart reveals that the 49 age−related metabolites mainly belong to four categories. (**C**) There is a total of 49 differential metabolites significantly correlated with age (the heatmap exclusively displays age-related metabolites). The size and color of each scatter plot point showed the *p* values of Spearman’s correlation and correlation coefficient values (red points represented positive correlations, *p* < 0.05 and R > 0.3; blue points represented negative correlations, *p* < 0.05 and R < −0.3). (**D**). The plot for the results of canonical pathway analysis. Nodes represent significantly enriched metabolic pathways. The x axis shows overlap rate of numbers actually matched from the user-uploaded metabolites and the total number of molecules in the pathways. The y axis shows the pathway names. (**E**) Heatmap showing the relative abundance of three groups. Only metabolites with significant differences between at least two groups and correlation with age are displayed (see Appendix A and Figure 3C). The numbers on the bottom represent ages. Yo, juvenile, *n* = 9; Ad, adult, *n* = 12; Ag, oldness, *n* = 15.

**Figure 4 microorganisms-11-02406-f004:**
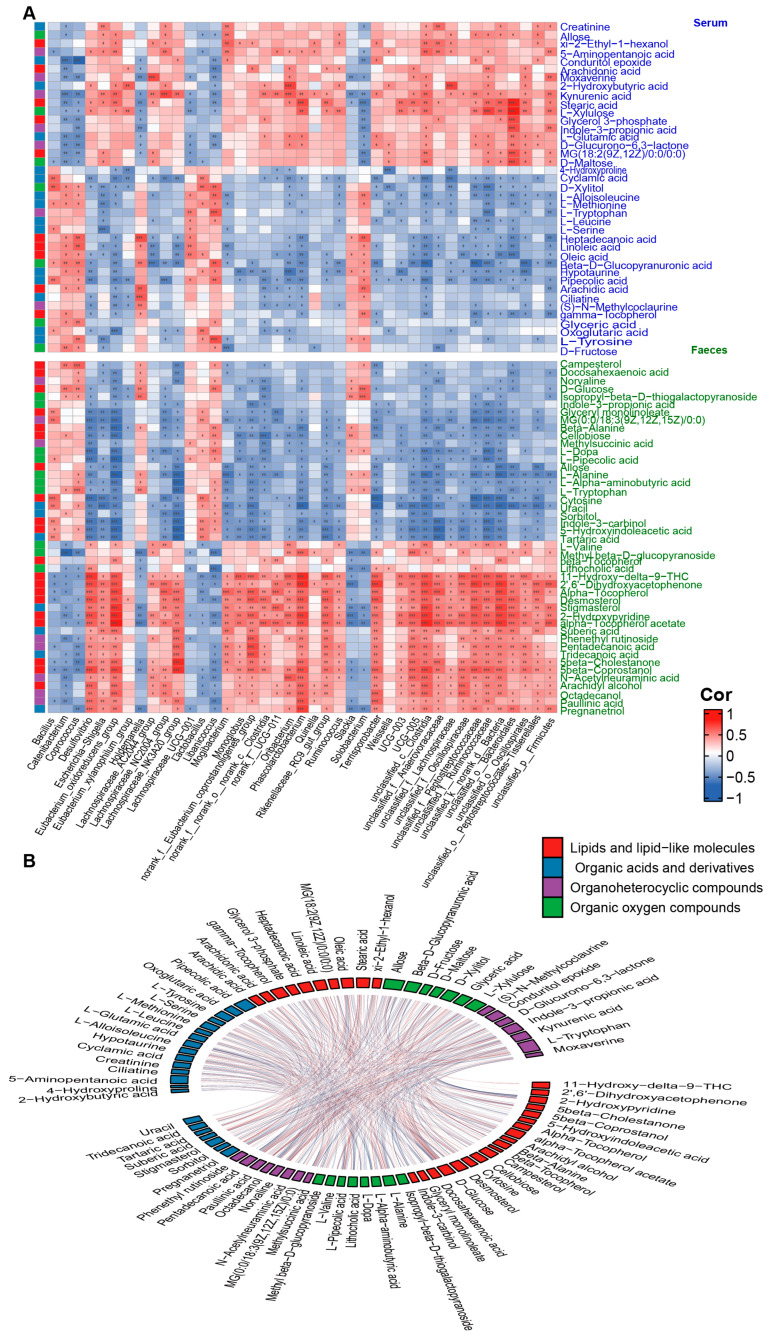
Correlation analysis of age−dependent factors. (**A**) Heat map of the Spearman’s rank correlation coefficient of gut microbial genera with organic acids and derivatives (blue legend), lipids and lipid-like molecules (red legend), organic oxygen compounds (green legend), and organoheterocyclic compounds (purple legend) in serum (blue font) and feces (green font). Red squares indicate positive correlation between genera and metabolites, green squares suggest negative correlation. The statistical significance is denoted on the squares (* *p* < 0.05; ** *p* < 0.01; *** *p* < 0.001). (**B**) A Chord diagram visualizing the significant interrelation between age-dependent serum metabolites (upper part) and fecal metabolites (lower part). Metabolites are classified into four categories: organic acids and derivatives (blue cells), lipids and lipid-like molecules (red cells), organic oxygen compounds (green cells), and organoheterocyclic compounds (purple cells). The red connectors represent significant positive correlations between two metabolites (*p* < 0.05 and R > 0.3), while the blue connectors indicate significant negative correlations between two metabolites (*p* < 0.05 and R < −0.3).

**Figure 5 microorganisms-11-02406-f005:**
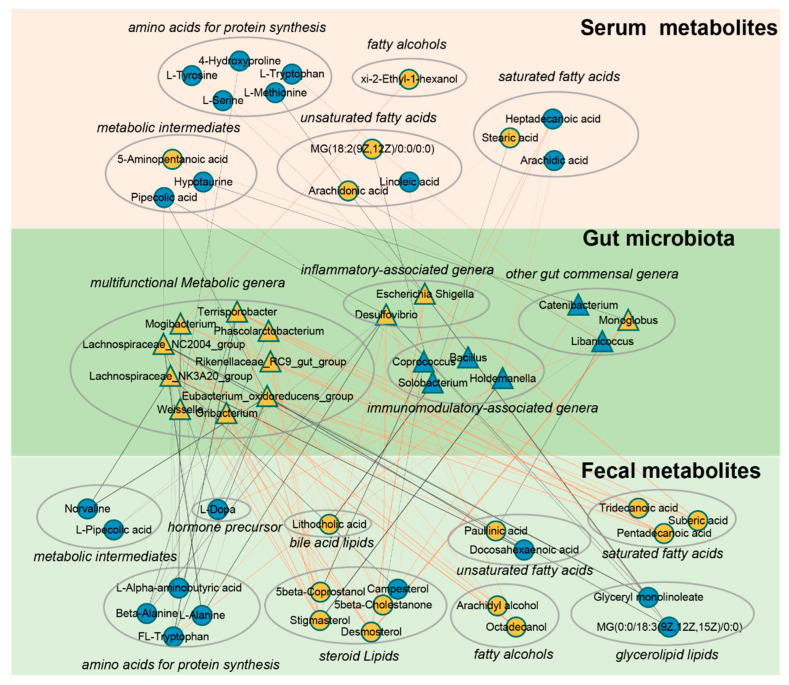
A co−occurrence network reflecting strong correlation of age−dependent gut microbes with host amino acids and lipids. The network based on strong correlation (*p* < 0.05, R > |0.5|) of age-dependent gut microbes with host amino acids and lipids. Triangles denote age-correlated gut genera, while circles represent age-associated host amino acids and lipids. Yellow triangles or circles indicate a significant positive correlation with age, whereas blue signifies a significant negative correlation. Lines between nodes indicate strong negative (light blue) or positive (light red) correlation (R > |0.5|), and line thickness indicates the *p*−value (*p* < 0.05).

## Data Availability

The raw data of 16s RNA sequencing has been uploaded to the public database (http://www.ncbi.nlm.nih.gov/bioproject/1020531, accessed on 20 September 2023) with the accession ID PRJNA1020539. The raw data of GC-MS has been uploaded to the public database CNCB (https://www.cncb.ac.cn/, accessed on 20 September 2023) with the accession id OMIX005001-01.

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
