# Peer review of "Multi-Omics Analysis Reveals Age-Related Microbial and Metabolite Alterations in Non-Human Primates"

_microorganisms, 2023, doi:10.3390/microorganisms11102406_

Round 1

Reviewer 1 Report

The reviewer highly appreciates the authors' multi-omics analysis to investigate the age-related microbial and metabolite alterations in non-human primates. However, there are few comments which the authors need to address; 

1.  Age stratification: The reviewer checked the references provide by the authors on age stratification (Li et al 2019 and Janiak et al 2021) for this study. Hence, the inclusion of 16yr rhesus macaques in old age category need to be addressed. In both the references, the author mentioned aged or old adults instead of adults for above 15 yr. 

Also, the reviewer would like to recommend authors to read the published article titled; "An overview of nonhuman primates in aging research" by Juile et al (DOI:10.1016/j.exger.2016.12.005)

In general, rhesus monkeys aged 1522 years are deemed middle aged, while those over 30 years are considered old or elderly" 

2. The reviewer would suggest the author to mention a sentence or a paragraph in introduction section regarding 16s rRNA. 

Author Response

For research article

Response to Reviewer 1 Comments

1. Summary

Many thanks for reviewing our manuscript and offering valuable constructive suggestions. According to these comments, we have revised the manuscript. The responses, explanations and revisions related to the suggestions were listed below. All traces of modification were also shown in tracked copy and clean copy.

2. Questions for General Evaluation

Reviewer’s Evaluation

Response and Revisions

Does the introduction provide sufficient background and include all relevant references?

Can be improved

Are all the cited references relevant to the research?

Yes

Is the research design appropriate?

Yes

Are the methods adequately described?

Yes

Are the results clearly presented?

Yes

Are the conclusions supported by the results?

Yes

3. Point-by-point response to Comments and Suggestions for Authors

Comments 1: Age stratification: The reviewer checked the references provide by the authors on age stratification (Li et al 2019 and Janiak et al 2021) for this study. Hence, the inclusion of 16yr rhesus macaques in old age category need to be addressed. In both the references, the author mentioned aged or old adults instead of adults for above 15 yr. Also the reviewer would like to recommend authors to read the published article titled; "An overview of nonhuman primates in aging research" by Juile et al (DOI:10.1016/j.exger.2016.12.005). "In general, rhesus monkeys aged 15–22 years are deemed middle aged, while those over 30 years are considered old or elderly".

Response 1: Thanks for your suggestions. We agree with the reviewer that stratification of aging stages in rhesus macaques are lack of consensus among research teams worldwide [1-5]. In this study, we used our previous stratification of aging stages in macaques [6,7] and also refer to the research from ML Li et al. [1] (please see lines 90 in clean copy). Therefore, we ultimately considered the age of ≥16 years as the old stage of rhesus macaques. Due to the lack of consensus in classifying old stages, we have added a discussion in the limitations section and cited an important study from Juile et al (DOI:10.1016/j.exger.2016.12.005). Please see lines 449-451 in clean copy: “The stratification of old stage in rhesus macaques has not yet reached consensus, which may be related to complex aging phenotypes observed in non-human primates [86]. Further studies are needed to unify the stratification in the future.

Comments 2: The reviewer would suggest the author to mention a sentence or a paragraph in introduction section regarding 16s rRNA.

Response 2: Thanks for your suggestions. We have added the details of 16S rRNA in the Introduction section. (please see lines 41-46 in clean copy). "16S rRNA gene sequencing, which amplifies and sequences specific regions of bacterial 16S rRNA genes, rapidly and efficiently identifies and categorizes microbial species in the gut microbiome. The technique provides a crucial tool for investigating the intimate relationship between the gut microbiome and host health, as well as the mechanisms behind various diseases[8]. However, there is still a need to elucidate the dynamic changes in gut microbiota during the aging process using 16S rRNA gene sequencing.”

  1. Li, M.L.; Wu, S.H.; Zhang, J.J.; Tian, H.Y.; Shao, Y.; Wang, Z.B.; Irwin, D.M.; Li, J.L.; Hu, X.T.; Wu, D.D. 547 transcriptomes from 44 brain areas reveal features of the aging brain in non-human primates. Genome biology 2019, 20, 258, doi:10.1186/s13059-019-1866-1.
  2. Rhoades, N.S.; Davies, M.; Lewis, S.A.; Cinco, I.R.; Kohama, S.G.; Bermudez, L.E.; Winthrop, K.L.; Fuss, C.; Mattison, J.A.; Spindel, E.R.; et al. Functional, transcriptional, and microbial shifts associated with healthy pulmonary aging in rhesus macaques. Cell reports 2022, 39, 110725, doi:10.1016/j.celrep.2022.110725.
  3. Beckman, D.; Ott, S.; Donis-Cox, K.; Janssen, W.G.; Bliss-Moreau, E.; Rudebeck, P.H.; Baxter, M.G.; Morrison, J.H. Oligomeric Aβ in the monkey brain impacts synaptic integrity and induces accelerated cortical aging. Proceedings of the National Academy of Sciences of the United States of America 2019, 116, 26239-26246, doi:10.1073/pnas.1902301116.
  4. Thomé, A.; Gray, D.T.; Erickson, C.A.; Lipa, P.; Barnes, C.A. Memory impairment in aged primates is associated with region-specific network dysfunction. Molecular psychiatry 2016, 21, 1257-1262, doi:10.1038/mp.2015.160.
  5. Keuker, J.I.; Luiten, P.G.; Fuchs, E. Capillary changes in hippocampal CA1 and CA3 areas of the aging rhesus monkey. Acta neuropathologica 2000, 100, 665-672, doi:10.1007/s004010000227.
  6. Duan, J.; Yin, B.; Li, W.; Chai, T.; Liang, W.; Huang, Y.; Tan, X.; Zheng, P.; Wu, J.; Li, Y.; et al. Age-related changes in microbial composition and function in cynomolgus macaques. Aging 2019, 11, 12080-12096, doi:10.18632/aging.102541.
  7. Tan, X.; Chai, T.; Duan, J.; Wu, J.; Zhang, H.; Li, Y.; Huang, Y.; Hu, X.; Zheng, P.; Song, J.; et al. Dynamic changes occur in the DNA gut virome of female cynomolgus macaques during aging. MicrobiologyOpen 2021, 10, e1186, doi:10.1002/mbo3.1186.
  8. Langille, M.G.; Zaneveld, J.; Caporaso, J.G.; McDonald, D.; Knights, D.; Reyes, J.A.; Clemente, J.C.; Burkepile, D.E.; Vega Thurber, R.L.; Knight, R.; et al. Predictive functional profiling of microbial communities using 16S rRNA marker gene sequences. Nature biotechnology 2013, 31, 814-821, doi:10.1038/nbt.2676.

Reviewer 2 Report

In the manuscript submitted to me for review entitled " Multi-omics reveals age-related microbial and metabolite alterations in non-human primatesthe authors Xiang Chen, Yiyun Liu, Juncai Pu, Siwen Gui, Dongfang Wang, Xiaogang Zhong, Wei Tao, Xiaopeng Chen, Weiyi Chen, Yue Chen, Renjie Qiao and Peng Xie present their research on age-related changes in the gut, serum and fecal metabolome in male rhesus macaques.

The research is extremely important because it helps to track some of the changes occurring in the body during the aging process. Such research would contribute to the control of these changes to control some of the symptoms accompanying the aging process and possibly increase life expectancy.

The methods used are correctly selected and described. The obtained results correspond to the conclusions drawn by the authors. The authors present their results in 5 figures, 6 supplementary figures and 1 supplementary table.

In support of their research, the authors presented 83 references, most of which are from the last 20 years, but there are also studies from 4 decades ago. Of all the references, 51 are from the last 5 years (nearly 2/3 of the total number), of which 5 are from 2023.

I congratulate the authors for the chosen topic and the scale of the research conducted. I have no objections to the conduct of the experiments themselves and their description. I have only a few recommendations that, if I were a reader of the article, I would have preferred to have been included in the manuscript by the authors.

1. The font in figures 1, 2, 3 and 4 is too small and illegible. It frustrates the reader. If possible, the authors could increase the font or the size of the figures themselves.

2. 6 supplementary figures and 1 table are presented in a separate file. A link to them is provided. But I personally am a reader who, when reading about some results, prefers to see them rather than looking for them somewhere else. I believe that the information in the supplementary figures is important to the results obtained, and my personal opinion is that at least some of the figures should be included in the main part of the manuscript.

3. Some of the references do not list all authors. I think it is more representative and complete when all authors are listed. These are reference numbers: 6, 9, 10, 12, 14, 17, 19, 21, 23, 25, 26, 30, 32, 33, 38, 39, 55, 57, 59, 64, 65, 73 and 79. Let all the authors be added.

Author Response

For research article

Response to Reviewer 2 Comments

1. Summary

Many thanks for reviewing our manuscript and offering valuable constructive suggestions. According to these comments, we have revised the manuscript. The responses, explanations and revisions related to the suggestions were listed below. All traces of modification were also shown in tracked copy and clean copy.

2. Questions for General Evaluation

Reviewer’s Evaluation

Response and Revisions

Does the introduction provide sufficient background and include all relevant references?

Yes

Are all the cited references relevant to the research?

Yes

Is the research design appropriate?

Yes

Are the methods adequately described?

Yes

Are the results clearly presented?

Can be improved

Are the conclusions supported by the results?

Yes

3. Point-by-point response to Comments and Suggestions for Authors

Comments 1: The font in figures 1, 2, 3 and 4 is too small and illegible. It frustrates the reader. If possible, the authors could increase the font or the size of the figures themselves.

Response 1: Thanks for your suggestions. We have revised all these figures. Please see the changes in tracked and clean copies.

Comments 2: 6 supplementary figures and 1 table are presented in a separate file. A link to them is provided. But I personally am a reader who, when reading about some results, prefers to see them rather than looking for them somewhere else. I believe that the information in the supplementary figures is important to the results obtained, and my personal opinion is that at least some of the figures should be included in the main part of the manuscript.

Response 2: Thanks for your constructive comments. Considering the importance of interpretation of the results, we have incorporated the Figure S6 into Figure 3. Please see the changes in tracked and clean copies.

Comments 3: Some of the references do not list all authors. I think it is more representative and complete when all authors are listed. These are reference numbers: 6, 9, 10, 12, 14, 17, 19, 21, 23, 25, 26, 30, 32, 33, 38, 39, 55, 57, 59, 64, 65, 73 and 79. Let all the authors be added.

Response 3: Thanks for your suggestions. We have rechecked all the references and added the complete names of all authors. Please see the changes in tracked and clean copies.